# How does Indian news media report smokeless tobacco control? A content analysis of the gutka ban enforcement

**Vivek Dsouza**[1], **Pratiksha Mohan Kembhavi**[1], **Praveen Rao S.**[1], **Kumaran P.**[1], **Pragati B. Hebbar**[1,2] *

**1** Chronic Conditions and Public Policies cluster, Institute of Public Health Bengaluru (IPH), Karnataka, India,
**2** Department of Health Promotion, Maastricht University (CAPHRI), Maastricht, The Netherlands

* pragati@iphindia.org

## Abstract

Smokeless tobacco (SLT) products like gutka and paan masala are a growing public health crisis in India. Despite enacting a ban—the highest form of regulation—little is known about implementation progress. The purpose of this study was to look at how enforcement of gutka ban is covered in Indian news media and if media is a reliable source of data. We conducted a content analysis of online news reports (n = 192) from 2011 to 2019. News characteristics such as name and type of publication, language, location, slant and beat coverage, visuals, and administrative focus were quantified. Similarly, news contents were inductively coded to examine dominant themes and the implementation landscape. We found that coverage was initially low but increased after 2016. Overall, news reports were in favor of the ban. Five leading English newspapers covered the majority of the ban enforcement reports. Prominent themes like consumption, health hazards, tobacco control responses, impact on livelihoods, and illicit trade were drawn from the textual analysis as the main arguments in relation to the ban. Gutka is largely seen as an issue of crime reflected by the contents, sources, and frequent use of pictures depicting law enforcement. The interconnected distribution channels of the gutka industry hindered enforcement, highlighting the need to study the complexities of regional and local SLT supply chains.

## Introduction

India is one of the largest producer and consumer of smokeless tobacco (SLT) products in the world, with twice the prevalence as compared to smoking cigarettes or bidis [1]. SLT—consumed in various forms—causes oral and esophageal cancers and increases the risk of hypertension and cardiovascular diseases [2, 3]. An estimated 200,000 people die each year due to SLT which costs the Indian economy INR 464.2 billion [4, 5]. In addition, long-term SLT use is driven by a myriad of socio-economic, cultural, and behavioral factors, and is more widespread in rural areas with limited healthcare options [6, 7]. Considering the detrimental health impact of SLTs, India enacted a number of international and domestic tobacco control laws to discourage consumption. In 2004, it ratified the WHO Framework Convention on Tobacco

**Data Availability Statement:** Data related to this study have been supplied in the supporting information materials.

**Funding:** This work was supported by the DBT/ Wellcome Trust India Alliance Early Career Fellowship [IA/CPHE/17/1/503338] awarded to PH. The funders had no role in study design, data collection and analysis, decision to publish, or preparation of the manuscript.

Control (WHO-FCTC), and since then, several best practices have been implemented [8]. More recently, state governments in India banned the sale of gutka and pan masala—popular SLT variants—using food laws as a result of multi-sectoral coordination between the Supreme court, policymakers, and NGOs [9, 10]. Despite concerted efforts, it is unclear how much of illicit gutka is widely available in the market and what is required to strengthen SLT control in India [11]. In such a case, the media can provide additional insight regarding the enforcement of the ban.

The media is a valuable resource that helps shape people's knowledge, beliefs, and attitudes by drawing attention to contemporary issues and encouraging critical debates around policy priorities and outcomes. As a health promotion strategy, public health advocates frequently use the media to communicate the risks of tobacco. Studies have shown that media campaigns focusing on pictorial health warnings effectively reduce the likelihood of tobacco use initiation and motivate people to quit [12–15]. Corporations, on the other hand, persuade media firms to omit facts and report uncritically. There is strong evidence of tobacco industries employing a number of tactics to undermine tobacco control legislations. These include corporate social responsibility initiatives (to cultivate a favorable image), aggressive marketing and promotion of new tobacco products (to retain customers and profits), and disseminating industry-sponsored research (to mislead the public), among others [16, 17]. According to a content analysis of US newspapers from 2006 to 2010, the media regularly downplayed SLT-related health hazards by referring to it as a 'less harmful' product [18]. This is alarming given the $576.1 million investment to increase the acceptability of SLT in regions where smoking is declining [19]. More importantly, the way non-communicable diseases are framed indicates that large-scale private sector companies are able to persuade the media to overlook the commercial determinants of health and shift the burden of responsibility to the user [20–23]. To safeguard business interests, tobacco industries employ similar tactics to influence how SLT is portrayed in the media by determining what is considered newsworthy and which sources are frequently cited [24].

In light of growing industry-media relations and increasing use of digital media services, how has Indian news media reported the enforcement of the gutka ban? As part of our ongoing study on the implementation of tobacco control policies in India [25], we aim to analyze patterns in the coverage of the ban and examine whether the media is a reliable source of enforcement-related data. The exploratory nature of the inquiry will allow us to better understand the gutka ban enforcement landscape, actors involved, and the facilitators and barriers that exist.

## Materials and methods

We conducted a media content analysis using Braun and Clarke's six-step method: 1) Data familiarization, 2) Generating initial codes, 3) Searching for themes, 4) Reviewing themes, 5) Defining and naming themes, and 6) Producing the final output [26].

### Search strategy

We used Google search engine database to search for news reports on the gutka ban. We used the search term 'gutka seizure' as it is one of the most commonly used enforcement strategies in India. A secondary search was conducted for each Indian state and union territory using the syntax ["gutka seizure" AND "Indian state/union territory" AND "state capital" AND "three highly populated cities"]. Since gutka was banned at different time periods in different Indian states, we did not apply restrictions by date. The searches were carried out in Bengaluru, the city capital of Karnataka state in southern India.

## Inclusion and exclusion criteria

We included online news reports published till the year 2019. News on enforcement of smoke-free policies were not included. Our search terms were developed with the intention of retrieving English news. However, we came across many Hindi and Marathi news reports during the initial search. We included them based on their high readership [27] and circulation [28], state-specific relevance, and the research team's time and ability to translate the language. We did not conduct specific or additional searches in Hindi and Marathi. Due to limitations in fully understanding, interpreting, and translating regional languages, we did not include news published in 20 other scheduled languages including Kannada—the state language of Karnataka. News reports with expired web addresses, behind paywalls and published on blogsites were also excluded. In terms of content (subject matter), we included news reports with and without images and excluded audio-visual and verbatim (exact duplicate) text. We included two or more reports on the same enforcement activity only if they offered any additional details about the incident.

## Selection and analysis of news reports

PR (having a background in media relations and advocacy) conducted the online searches and compiled the news reports in a Microsoft Word document. Next, VD (trained in interdisciplinary social sciences) and PH (experienced in tobacco control and health policy and systems research) independently assessed the news as per their eligibility; any disagreements were resolved through team discussions. To familiarize with the data, PK (early career public health researcher) and VD read and re-read each eligible news report and took handwritten notes that were later developed into twelve categories. Descriptions for each category were curated and refined with the help of KP (having a background in journalism) and PH using an online resource manual as shown in Table 1 [29]. Descriptive analysis was used to quantify and analyze the news characteristics.

**Table 1. Data extraction and coding schema.**

| Category | Description |
|---|---|
| Year | Year of publication. |
| Name of publication | Name of the publishing house or groups. |
| News type | Classification of news media such as newspaper, news agency, news magazine, financial magazine, news portal, news media company. |
| News content | News article or News analysis. |
| Language | Language in which news is reported. |
| By-line | The name of the writer and/or group mentioned in the story. |
| Location | State or union territory where the news is reported. |
| Slant | |
| • Positive | Supportive of the ban |
| • Neutral | Neither supportive nor opposing the ban but presents a mixed review |
| • Negative | Opposing or against the ban |
| Beat | A subject matter that a reporter frequently covers. |
| Administrative focus | Level at which enforcement is reported (national, state, district/city). |
| Visuals | Pictures used to convey the story |
| Enforcement timing | Stage in the gutka supply chain at which enforcement is reported. |

For textual analysis, PK, PR, and KP extracted preliminary data in a Microsoft Excel spread sheet. PH and VD reviewed a 10% sample and differences were resolved with the team. Next, the news reports were imported into NVivo software (version 12). Open coding format was applied where PK and VD inductively coded the contents of the news reports while PH reviewed the themes and provided feedback for improvement. Differences were addressed through multiple discussions with PR and KP.

## Results

We retrieved 173 news reports during the initial search and 55 reports during the second-level search. After screening and removing duplicates, we evaluated 226 reports by reading their headlines (title), and eight were found to be unrelated, leaving 218 reports. Of these, we assessed the news contents and excluded 26 based on the eligibility criteria, resulting in 192 reports for the review as depicted in Fig 1.

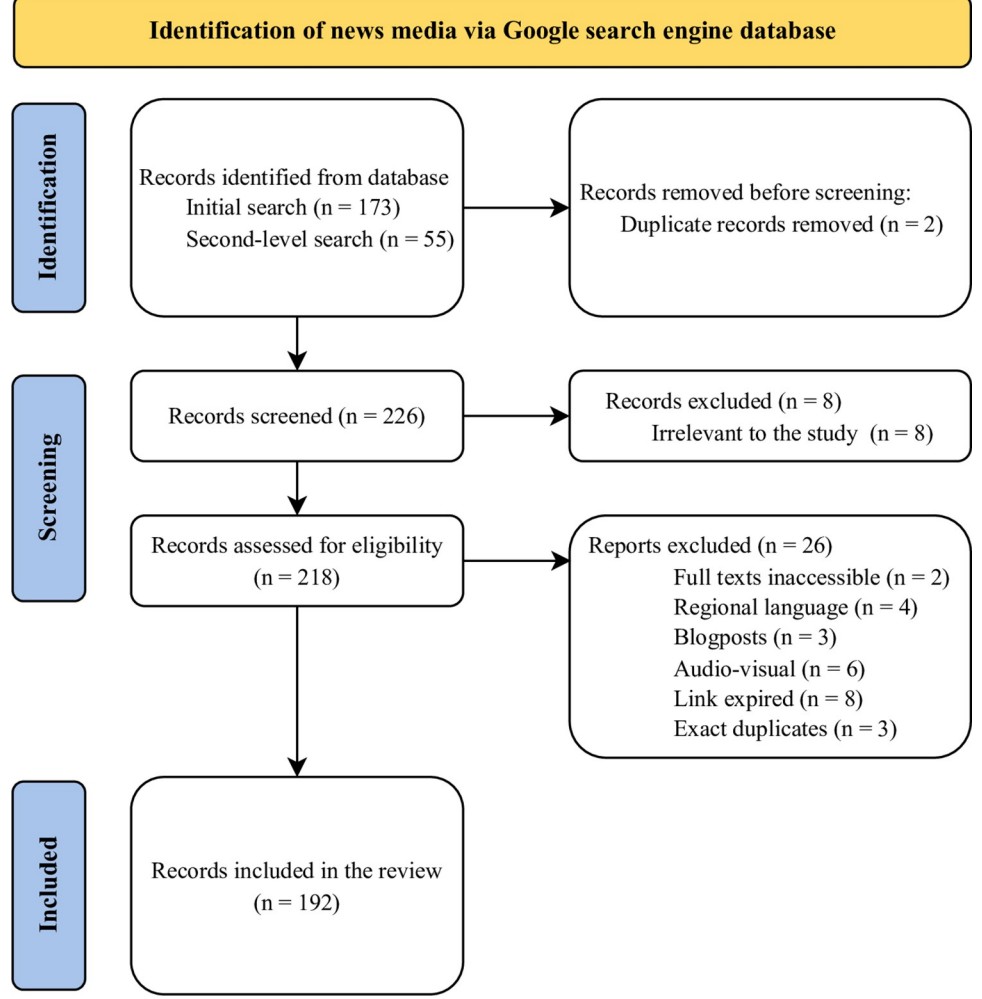

**Fig 1. Steps involved in identifying, screening, and assessing the eligibility of news reports.**

## Characteristics of the news reports

Table 2 describes the news characteristics that met our eligibility criteria. Detailed news characteristics are included in S1 and S2 Tables. Among the news reports studied, news articles (n = 152) were written using the *who, what, when, where, and why* format and news analysis pieces (n = 40) provided additional information regarding the ban enforcement.

## Major themes covered

**Consumption behavior.** Gutka use experimentation and initiation were highest among young people. Despite the ban, news reports expressed concerns regarding over consumption, dual use, and switching to twin packets (pan masala and flavored tobacco). Increase in prices or scarcity of the products had little or no effect on users as they were willing to pay higher prices for it.

"*Post-ban on the production and sale of gutka in Gujarat since September 11, 2012, entire 100 per cent of the gutka consumers have switched over to something similar to gutka by buying ingredients separately to combine with tobacco and consumed as gutka or a product similar to gutka. Also, 43 to 55 percent consumers had initiated gutka below 20 years of age.*"—The Indian Express, 2015 (055)

"*A study released by Adyar Cancer Institute in 2017, based on interviews with over a lakh people, found that more than 90% of smokeless tobacco users in the state have no difficulty in procuring banned gutka products, although they pay double the price to buy it.*"—The Times of India, 2018 (152)

**Health and environmental hazards.** Addiction among adolescents was the most frequently reported health concern. This was because of the product's accessibility in the market. Due to the reliance on gutka, companies began selling adulterated products, often laced with toxic substances, endangering both food safety and health. Oral cancers and other tobacco-related diseases were also reported. One report identified plastic waste caused by discarded gutka packets as having a negative impact on the environment.

"*According to a police source, collector K Rajamani had conducted a meeting with all the department heads three weeks ago when the education department officials brought his attention to the sale of banned tobacco products, especially Cool Lip, among the student community. 'The officials said a section of school students had become addicted to tobacco products and they were attending classes after consuming the same. They also expressed concerns over the fate of the student community.'*"—The Times of India, 2019 (204)

"*These tobacco product sellers distribute the outdated tobacco products to various retail shops of the district and as a result people's health is put at risk. Youths and college students mostly consume these tobacco products and are at a risk of suffering from oral diseases and even cancer.*"—The Sentinel, 2017 (024)

"*So they are also preserving the packets which is giving scope for adulteration and mixing with some harmful products. Patients are also facing mouth, throat, and tract related health diseases due to the dangerous substances.*"—The Hans India, 2017 (021)

**Response of tobacco control and activism.** Four reports mentioned different tobacco control tactics. To raise awareness about gutka-related health hazards, the local police issued public service announcements and held awareness drives for students, as well as for shops,

**Table 2. Key characteristics of the eligible news reports.**

| Category | Description | Total (N) | % |
|---|---|---|---|
| Year of publication | | | |
| | 2019 | 79 | 41.1 |
| | 2018 | 69 | 35.9 |
| | 2017 | 16 | 8.3 |
| | 2016 | 4 | 2.1 |
| | 2015 | 6 | 3.1 |
| | 2014 | 3 | 1.6 |
| | 2013 | 7 | 3.6 |
| | 2012 | 6 | 3.1 |
| | 2011 | 2 | 1.0 |
| Name of publication | | | |
| | **Publications with >10 reports** | **116** | **60.4** |
| | Times of India | 49 | 25.5 |
| | The Hans India | 27 | 14.1 |
| | Deccan Chronicle | 14 | 7.3 |
| | The Hindu | 13 | 6.8 |
| | The New Indian Express | 13 | 6.8 |
| | **Publications with <10 reports** | **76** | **39.6** |
| News type | | | |
| | Newspaper | 165 | 85.9 |
| | News Portal | 16 | 8.3 |
| | News Agency | 4 | 2.1 |
| | News media Company | 4 | 2.1 |
| | News Magazine | 2 | 1.0 |
| | Financial Magazine | 1 | 0.5 |
| News content | | | |
| | News Article | 152 | 79.2 |
| | News Analysis | 40 | 20.8 |
| Language | | | |
| | English | 166 | 86.5 |
| | Hindi | 19 | 9.9 |
| | Marathi | 7 | 3.6 |
| By-line | | | |
| | Bureau | 130 | 67.7 |
| | Writers' names | 62 | 32.3 |
| Location | | | |
| | **States with >10 coverage** | **156** | **81.3** |
| | Tamil Nadu | 52 | 27.1 |
| | Telangana | 39 | 20.3 |
| | Maharashtra | 35 | 18.2 |
| | Andhra Pradesh | 18 | 9.4 |
| | Chhattisgarh | 12 | 6.3 |
| | **States with <10 coverage** | **36** | **18.8** |
| Slant | | | |
| | Positive | 171 | 89.1 |
| | Negative | - | - |
| | Neutral | 21 | 10.9 |

*(Continued)*

**Table 2.** (Continued)

| Category | Description | Total (N) | % |
|---|---|---|---|
| Beat | | | |
| | Crime | 156 | 81.3 |
| | City and neighbourhood | 27 | 14.1 |
| | City and crime | 6 | 3.1 |
| | Health | 2 | 1.0 |
| | Others | 1 | 0.5 |
| Administrative focus | | | |
| | State | 54 | 28.1 |
| | District/city | 138 | 71.9 |
| Visuals | | | |
| | **No picture** | **55** | **28.6** |
| | **Representational** | **44** | **22.9** |
| | Consuming gutka | 17 | 8.9 |
| | Handcuffs | 13 | 6.8 |
| | Law enforcement | 6 | 3.1 |
| | Warning sign | 3 | 1.6 |
| | Other tobacco products | 2 | 1.0 |
| | Jail | 1 | 0.5 |
| | Ganja | 1 | 0.5 |
| | City | 1 | 0.5 |
| | **Enforcement** | **67** | **34.9** |
| | Enforcement team with seized gutka products | 28 | 14.6 |
| | Enforcement team with offenders and seized gutka products | 10 | 5.2 |
| | Only seized gutka products | 15 | 7.8 |
| | Enforcement activity | 8 | 4.2 |
| | Enforcement team with the offenders | 2 | 1.0 |
| | Offenders and seized gutka products | 1 | 0.5 |
| | Offenders | 1 | 0.5 |
| | Seized money | 1 | 0.5 |
| | Seized ganja products | 1 | 0.5 |
| | **Others** | **26** | **13.5** |
| | Gutka products on display | 16 | 8.3 |
| | Pan shop | 6 | 3.1 |
| | Gutka manufacturing unit | 2 | 1.0 |
| | Image of the victim | 1 | 0.5 |
| | Google map | 1 | 0.5 |
| Enforcement timing | | | |
| | Transporting | 78 | 40.6 |
| | Storing | 51 | 26.6 |
| | Selling | 31 | 16.1 |
| | Manufacturing | 11 | 5.7 |
| | Unassigned | 10 | 5.2 |
| | Not applicable | 11 | 5.7 |

during raids. Civil society acted as watchdogs, inquiring about tobacco product sales and the enforcement status under the Right to Information (RTI) act. They contributed by either

supporting or intervening when enforcement was lax. In one case, local activists called for the boycott of gutka by burning the seized products.

> "*Officials said the screening of retail and wholesale shops will continue, but four teams of 25 officials each have also been asked to search houses connected to those in the business. 'We are now educating food business operators, including 56 wholesalers in Chennai district. The raids will continue daily,' said S. Lakshmi Narayanan, designated officer of the department. A chunk of the wholesale dealers operate from Sowcarpet and T. Nagar. An official on one of the teams said that while retailers have taken the products off their shelves, they continue to sell them clandestinely. 'We have to educate proprietors at all such outlets,' he said.*"—The Hindu, 2013 (158)

> "*However, when an activist in Coimbatore filed an RTI to source the action taken details, his intention wasn't to find out how many retailers were prosecuted. "They are only the small fish, I wanted to find out how many manufacturers have been nailed. The reply said none.*"— The Times of India, 2018 (152)

> "*KMSS activists forcibly seized tobacco, gutka, and flavoured pan masala worth lakhs of rupees from a Tinsukia-based trader and burnt it to ashes on Tuesday. KMSS assistant general secretary (tinsukia district) said, "We have taken this extreme step after the administration failed to stop their sale, despite the products being banned in the state.*"—The Telegraph, 2017 (023)

**Impact on livelihoods.**   Poverty was a deciding factor in entering the industry as shopkeepers would earn a living by selling gutka, pan masala, and other tobacco products. Despite public support, news reports expressed caution that the ban would negatively impact livelihoods and lead to unintended consequences. Without any alternatives, shopkeepers were forced to engage in illegal activities to compensate for their losses. Revenue losses at the national level were also attributed to the ban.

> "*Pan shops in various parts of the city are now engaged in either directly or indirectly abetting other illegal activities to make up for the losses incurred due to the crackdown. Some pan shops at Charminar, Kalapather, Golconda, Bandlaguda, Katedan, Rajendran-agar, Bhavaninagar, Afzalsagar and Shaheenagar are now selling packs of playing cards. A few months ago, the police also caught the owners of a few pan shops accepting cricket betting amounts and satta amounts from bookies. In the long run, almost all pan shops will turn into hotspots or facilitation points for illegal activities. A long-term economic assistance plan should simultaneously be drawn up for them, otherwise the crackdown will result in new challenges.*"—Deccan Chronicle, 2017 (006)

> "*Due to poverty and other reasons, these petty traders go to other districts particularly West Godavari, East Godavari and Guntur and buy the gutka.*"—The Hans India, 2018 (016)

> "*The ban is expected to result in a revenue loss of Rs. 100 crore per year to the exchequer.*"— Business Standard, 2012 (199)

**Illicit gutka trade.**   News reports expressed concerns regarding the unabated sale of gutka in several retail outlets. One of the reasons was tax evasion through smuggling, in which illegal traders smuggled gutka and other tobacco products from neighboring states and sold in the black market. These products did not have health warnings or manufacturer's details. Because the smuggled gutka is sold at higher premiums, the money that went into the black market was used for a variety of criminal activities. Five news reports citied the involvement of top government officials and industry executives in a multi-crore gutka scam.

"*Despite ban on the use of tobacco products and gutkha under the Assam Bill which was enforced as an Act on 2014 in the State, a section of black marketers are allegedly selling gutkha and other tobacco products in Assam. Similarly, in Golaghat district in various places a black market of tobacco products is flourishing.*"—The Sentinel, 2017 (024)

"*One of the reasons for the thriving black market for tobacco products is that they are easily smuggled in small quantities and sold at a big premium.*"—Mirror Now News, 2019 (148)

"*The gutka scam came to light on July 8, 2017, when Income Tax sleuths raided offices and residences of Jayam Industries (now known as Annamalai Industries) which had been facing charges of tax evasion worth Rs 250 crore and seized a diary containing incriminating evidence linking top officials of the state.*"—The New Indian Express, 2018 (139)

## Gutka ban enforcement landscape

**Stakeholders.** Individual stakeholders and those representing institutions at the national, sub-national, and local level were reported as being involved in formal policymaking and implementation of the ban as shown in Table 3.

**Table 3. Stakeholders and their roles in gutka ban enforcement as reported in the media.**

| Stakeholder | Role(s) |
|---|---|
| **Judiciary** (n = 8) | |
| • Supreme court | Policymaking, resolving disputes, and enforcing norms |
| • High courts | |
| **Government** (n=11) | |
| • Union and state governments | Policymaking, regulation and enforcement in public health |
| • District administration | |
| • Municipal corporations, *nagarpalika* | |
| **Central intelligence and investigation institutions** (n = 11) | |
| • Central Bureau of Investigation (CBI) | Investigating corruption |
| • Directorate General of GST Intelligence (DGGI) | Fighting tax evasion |
| • Narcotics Control Bureau | Combating drug trafficking |
| **State departments** (n = 10) | |
| • Health | Prevention, control, and management of diseases |
| • State tobacco control cell | Implementing provisions of COTPA[*] |
| • Commercial tax | Collecting taxes |
| • Food civil supplies and consumer protection | Enforcing orders passed under Essential Commodities Act, 1955 |
| **Regulatory authorities** (n = 62) | |
| • Food Safety and Standards Authority of India (FSSAI) | Regulating food safety |
| • Food and Drug Administration (FDA) | Enforcing legislation, testing samples |
| **Police** (n = 152) | |
| • Criminal Investigation Department | Investigating crimes |
| • Local police, special operations team, task forces, squads, railway protection force, local crime branch, detectives, rapid response (RR) teams, *mukhbir* or 'informant' | Maintaining law and order, public safety, enforcing laws, detecting and preventing criminal activities |
| **Academia and NGOs** (n = 4) | Tobacco control research and advocacy |

*The Cigarettes and Other Tobacco Products (Prohibition of Advertisement and Regulation of Trade and Commerce, Production, Supply and Distribution) Act, 2003

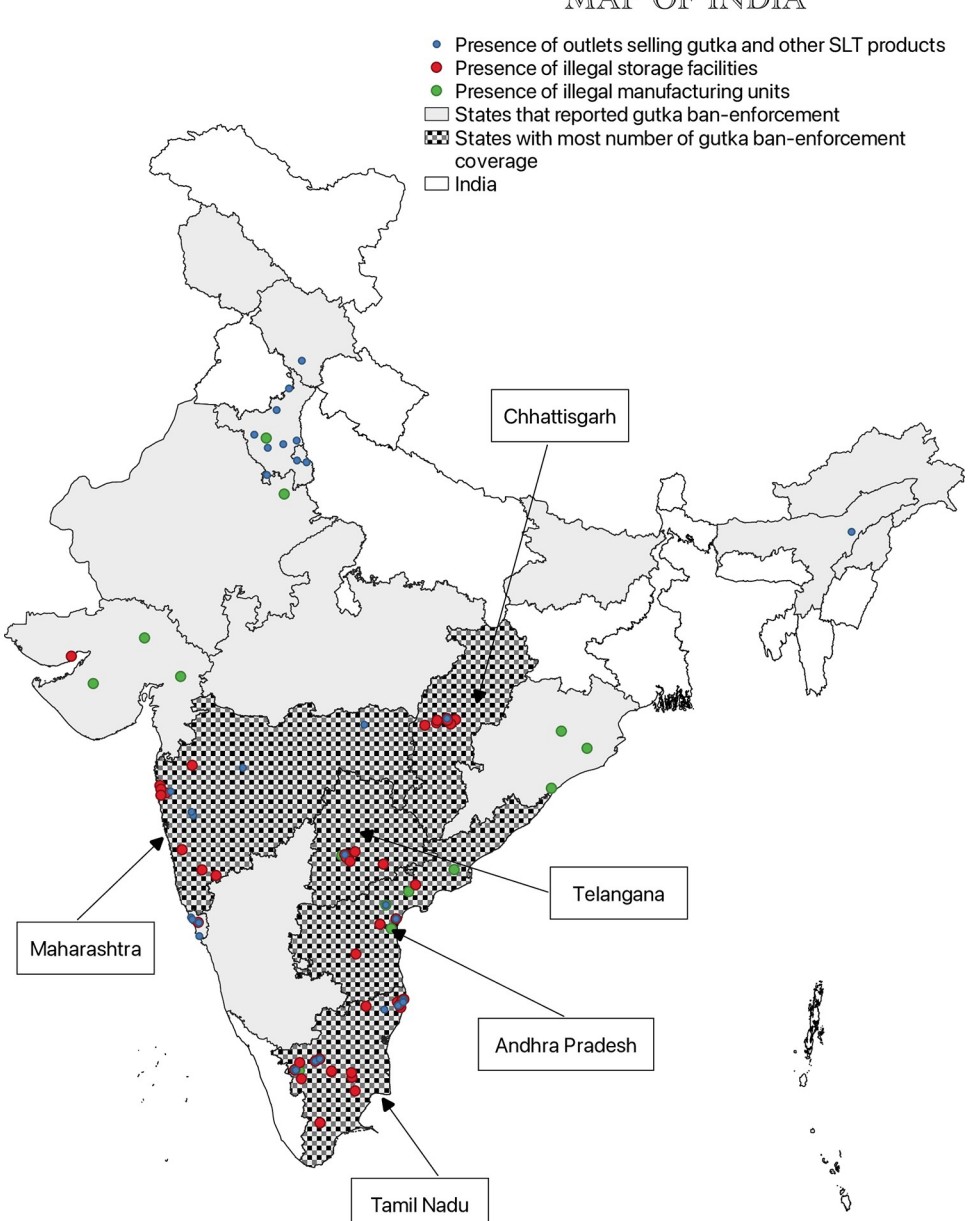

**Fig 2. Gutka seizures at manufacture units, storage facilities, and shops as reported in the media.** State Boundaries Maps are provided by Data{Meet} Community Maps Project. It is made available under the Creative Commons Attribution 2.5 India.

**Quantity and value of seized gutka.** Between 2011 and 2019, local police in coordination with agencies like the FSSAI, FDA, and DGGI enforced the ban at different locations and at multiple levels of the supply chain including facilities where gutka is produced and consumed. These were plotted on QGIS software, a geographic information system application, using location-based data reported in the news (Figs 2 and 3). The base map (shapefile) of the Indian state boundaries was downloaded from 'Community Created Maps of India,' a project run by the {DataMeet} community. During seizure, gutka worth lakhs of rupees in sachets as well as

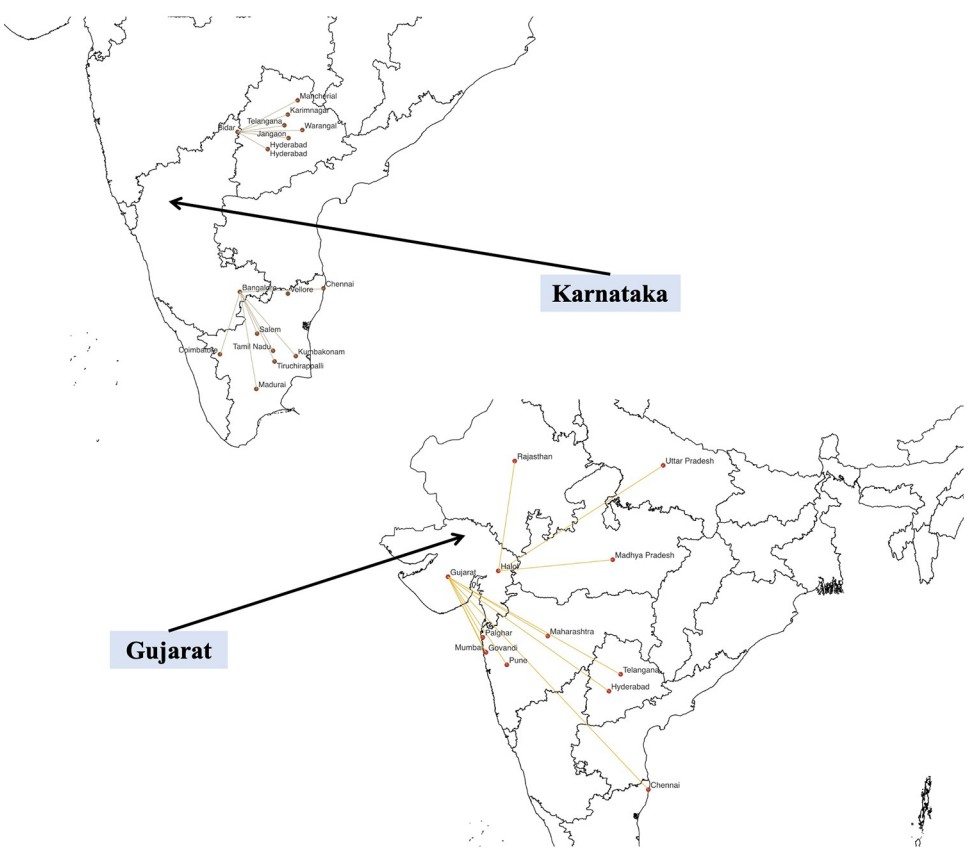

**Fig 3. Illicit transport of gutka in Karnataka and Gujarat as reported in the media.** State Boundaries Maps are provided by Data{Meet} Community Maps Project. It is made available under the Creative Commons Attribution 2.5 India.

bags containing crores-worth of gutka were recovered. Maharashtra collected the most fines totaling to ~ INR 561.28 crore followed by Telangana (~ INR 11 crore), and Tamil Nadu, Andhra Pradesh and Chhattisgarh (~ INR 4 crore each). Karnataka topped the list with a single-day seizure valued at ~ INR 400 crore. The quantity and value of gutka and other items recovered from different states is illustrated in S3 Table.

**Motivations and barriers.** Important motivational factors to the ban enforcement included champions, tip-offs, high-level support (orders), media scrutiny, and cash incentives (rewards). Barriers included ambiguity in the laws, contested viewpoints on gutka as a product, poor coordination, litigation tactics, and lack of resources which hindered enforcement. Table 4 represents a summary of motivations and barriers accompanied by the relevant quotations reported in the media.

**Gutka supply-chain network.** Four major supply-chain functions were identified based on the sequence of events described in the news. These were *procurement* of raw tobacco ingredients and finished products, *processing* (manufacturing and packaging of gutka), *logistics* (storing, transporting, and supplying), and *selling* (to vendors, shopkeepers and users). Key industry players (partners or associates, traders, wholesale distributors, retail vendors, merchants, shopkeepers, intermediaries like agents, brokers, and salesmen, supporting industries such as transport companies) were involved in the gutka supply chain—all of whom were connected by direct and indirect networks as shown in Fig 4.

**Table 4. Motivations and barriers while enforcing gutka ban in India.**

| Motivations | Quotes |
|---|---|
| Champion's role in enforcement (n = 88) | "*Two persons have been detained so far. The sample of the seized products has been sent for testing. Further investigation is on.*" Said the Durg city superintendent of Police, who is leading the campaign against the intoxicating drugs and the prohibited tobacco-based products." — The New Indian Express, 2019 (032) |
| Tip-offs (n = 51)<br>• Credible sources<br>• Informants hired by the police | "*K.A. Special team of Superintendent of Police got information that banned gutka was going from Khamgaon to Nagpur in truck number 16C-1780.*" — Lokmat, 2019 (096) |
| Orders/directives issued by higher authorities (n = 14)<br>• Union/state governments<br>• Food safety department<br>• Revenue department | "*On Wednesday, the state government issued directives to all government agencies—police, district collectors, and civic bodies—to implement the order. The directives were issued following a Supreme Court decision to allow the state to take action against offenders under section 328 of the Indian Penal Code (IPC).*" — Hindustan Times, 2018 (089) |
| Negative image of the police in the media (n = 5) | "*After the two seizures came under intense media scrutiny, Pon Manickavel, inspector-general of police Railways, directed the state railway police to take charge of the confiscated goods and to register an FIR.*" — The Times of India, 2018 (143) |
| Rewards (n = 2) | "*Superintendent of Police appreciated Gudihathnoor police and assured of presenting rewards to them.*" — The Hans India, 2019 (209) |
| **Barriers** | **Quotes** |
| Legislative and organizational issues<br>• Lack of clarity in the act<br>• Loopholes in the rules<br>• Poor coordination | "*The officials also suggest that the central government remove chewable tobacco from the purview of COTPA and bring it under FSSAI as per court order. The grey areas in the acts give freehand to the manufacturers of the stuff like chewable tobacco.*" — The Times of India, 2019 (125) |
| Lack of resources<br>• Human and financial resources<br>• Lack of infrastructure | "*The FDA office near ESIS Hospital premises in Wagle Industrial Estate has already become a virtual godown to all seized goods with no room to move around.*" — Moneylife India, 2012 (100) |
| Litigation tactics (n = 1) | "*Challenging the FDA resolution that imposed a complete prohibition on the transportation of finished products—pan masala and chewing tobacco—through the state even if manufactured and meant to be sold outside, the petitioner stated that they being transporters, only carried the goods which were not banned in many other states.*" — The Times of India, 2019 (082) |

## Discussion

SLT is a public health threat in low- and middle-income countries (LMICs), with changing consumption patterns and an increase in dual tobacco use [30]. In complex and fragmented systems with organizational, political, and information constraints, immediate action is required to identify and generate context-specific (local) evidence to inform and strengthen SLT control. We attempt to do this through a content analysis of online news sources to determine how gutka ban enforcement is covered in Indian media. Unlike tobacco control laws such as COTPA, which have national and state-level review systems for the prohibition of smoking, there is limited knowledge on the monitoring, reviewing, and supervision of SLT control, partly due to inconsistent enforcement and a lack of publicly available datasets [31]. We used enforcement data generated by the media stories to fill this knowledge gap as the news are accessible and are reported in 'real-world' settings. To our knowledge, this is the first media content analysis studying the *enforcement* component of SLT control in the LMIC context. Earlier content analyses were conducted in high-income countries (HICs) and studied SLT in relation to advertising and promotion among youth and adults in print and social media [32–37]. The main findings of our study showed that gutka ban enforcement is communicated to the public using a variety of arguments and visuals embedded in the actors, processes, and outcomes as shown in Fig 5.

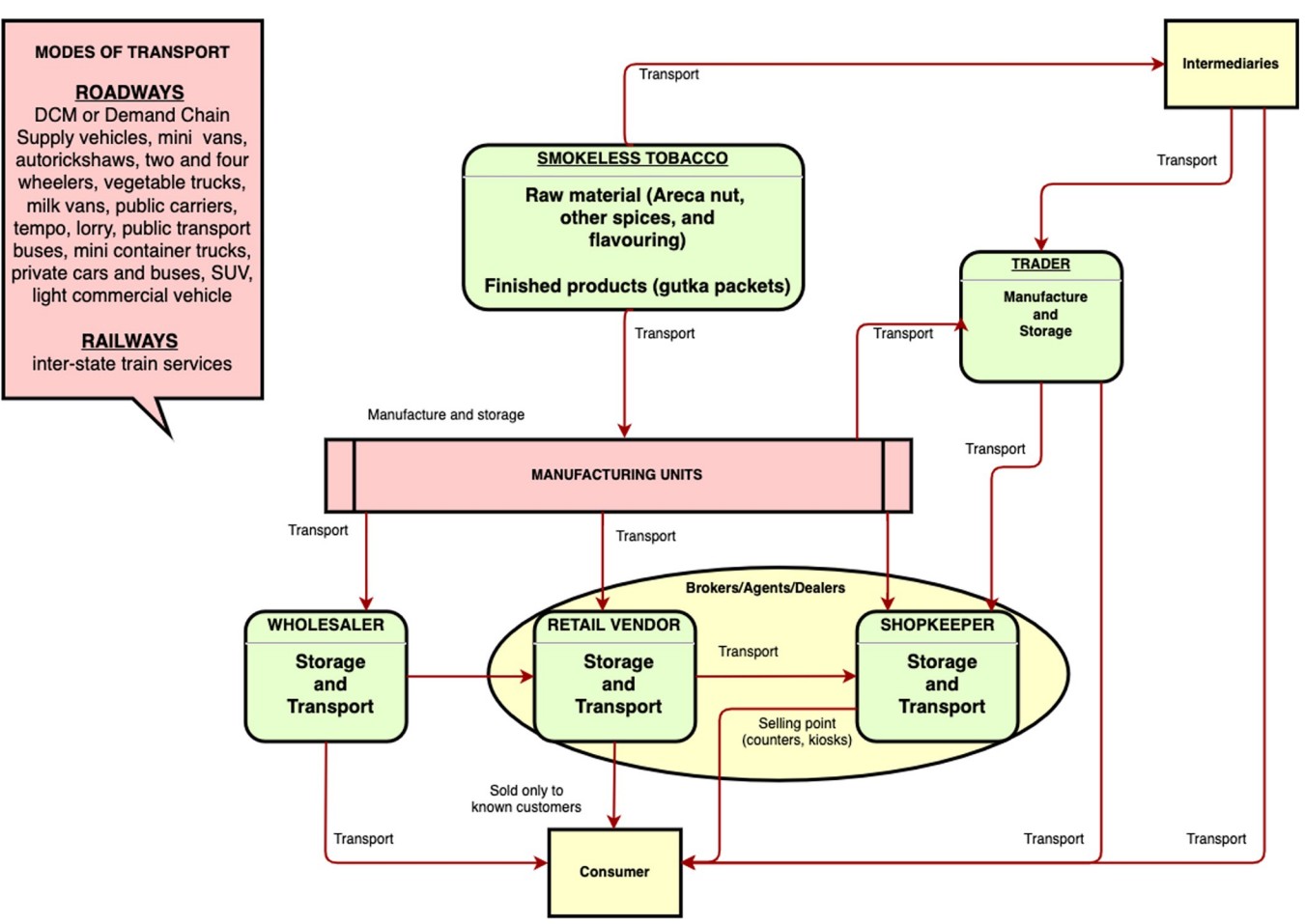

**Fig 4. Gutka supply-chain linking people, processes, and infrastructure to end users in India.**

The frequency of news coverage typically increases during major policy developments. In our case, however, there was low enforcement coverage since the enactment of the first ban (March 31, 2012). During the time period studied (2011 to 2019), there was a sharp increase in coverage after 2016. This was most likely due to an increase in the number of states passing the ban, as well as FSSAI's directive urging all states to strictly comply with the Supreme Court order [38]. In India, public health policies and programmes are designed at the national and sub-national levels while implementation occurs at the district and city level, which partially explains why most of the news (n = 138) focused on local-level enforcement. However, in terms of circulation, five English newspapers—The Times of India, The Hans India, Deccan Express, The Hindu, and The New Indian Express—having national and regional focus published the reports. This could be indicative of the small number of local news media included in our study. That being said, other factors like the proliferation of regional media outlets, the role of bureaus (n = 130) and news agencies in gathering and disseminating news remotely, and a lack of local correspondents could equally play a role [39].

Overall, the majority of the reports (n = 171) were in favor of the ban albeit for different reasons. Most of the news covered issues around health and illicit trade, which were the primary arguments for supporting the ban. A strong anti-tobacco culture (on health grounds) also appeared to have bolstered support. Some reports, although few in number, addressed the

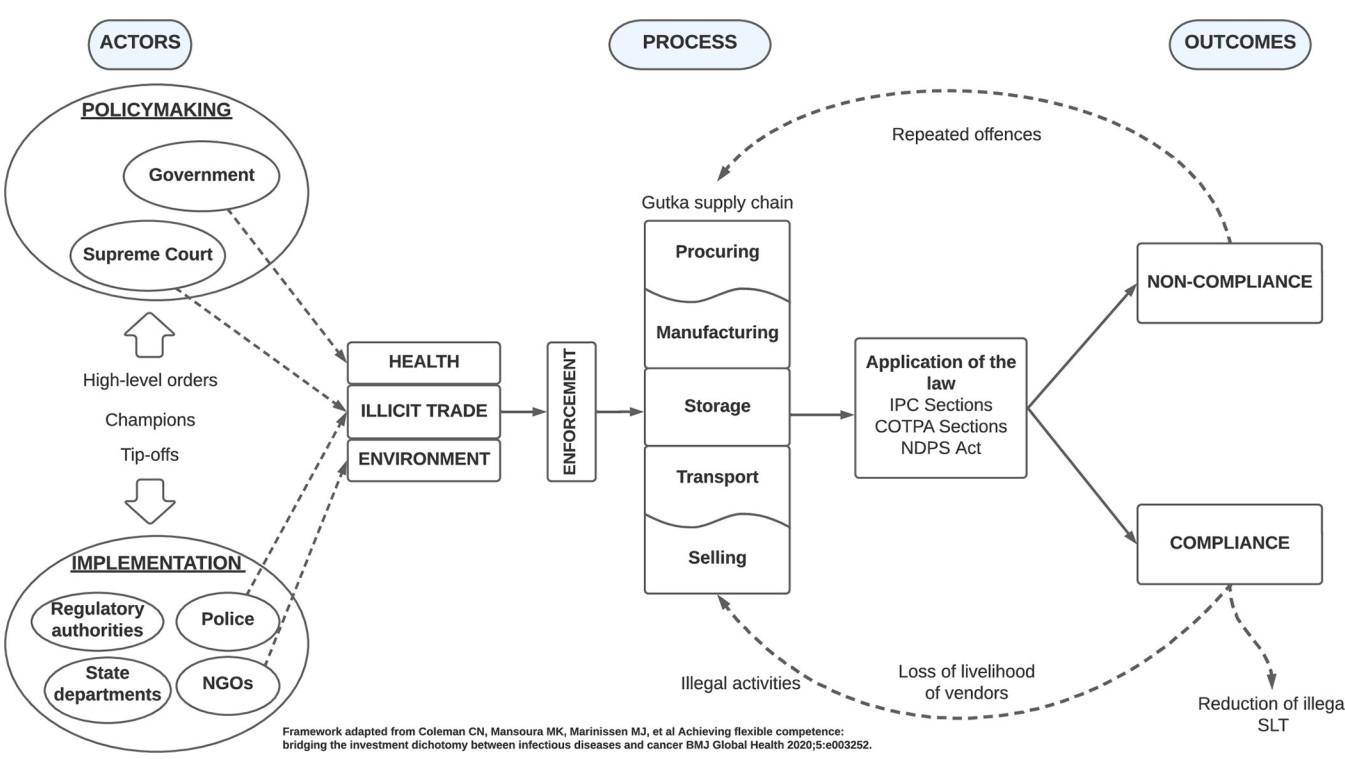

**Fig 5. How does Indian news media report gutka ban enforcement?**

ban's negative impact on the livelihoods of tobacco vendors. Despite health being central to the news, we observed that public health experts were rarely interviewed. On the contrary, most reports considered law enforcement agencies to be reliable and established sources of information, creating an impression that the police—a powerful figure and a symbol of the state—are largely responsible for protecting and promoting public health. Our claim is further supported by the frequent use of images depicting handcuffs, arrests, and prisons, as well as a strong preference for reporting news under the crime beat, which suggests that gutka is seen more as an individual-level criminal offence than a public health concern (which is a result of larger social and structural determinants) [40]. Our findings are similar to an Australian study on reporting practices, which found that when faced with resource and time constraints, writers rely on trusted sources and tend to be generalists, which impacts the quality and newsworthiness of a particular issue [41].

Several health and non-health sectors were involved in gutka ban enforcement. Among the stakeholders identified, police and regulatory authorities were well-represented in the news. We observed that in circumstances where enforcement was difficult, the police leveraged contacts with the CBI, FDA, and education departments to jointly enforce the ban [42]. Likewise, enforcement was dynamic when champions led the inspection drives and there was access to information (tip-offs) and high-level support. Rewards in the form of cash (n = 2) were also a motivating factor, which relate to two studies arguing that the effect of enforcement is stronger when officials are compensated by the fines collected [43, 44]. Several barriers raised by the news are consistent with global and Indian tobacco control literature, particularly with regard to gutka, where legal loopholes, lack of awareness and enforcement, and litigation by tobacco industries are associated with violation of the ban, sale of twin packets, without mandated health warnings [31, 45, 46]. Only four reports mentioned the role of academia or NGOs, in

either individual or institutional capacity, implying they are also involved but receive insufficient coverage. In terms of seizures, we found news media to be a useful resource in documenting the quantity and value of illicit gutka and other tobacco products. However, the data offered limited insight on state-wise implementation due to a range of SLT products in the market, a lack of disaggregated data, different units of measurement applied, and limited reporting capacities. To ensure data representation and validity, formal data collection and reporting systems need to be established and institutionalized so that comparative studies can determine which states are making progress and which are left behind.

We observed a growing presence of illegal gutka businesses in western, central, and southern India. Many storage facilities were reported in Gujarat and Tamil Nadu, which have a strong presence of chewing tobacco cultivation [47, 48]. Raw materials and finished products were transported from Gujarat to neighboring states like Maharashtra, Telangana Uttar Pradesh and Madhya Pradesh. However, several gutka manufacturing and storage units were also reported in Andhra Pradesh, dominated by FCV tobacco (used in cigarettes) which we did not expect (Fig 2). As chewing tobacco is not grown in Karnataka, it has likely emerged as a transit state for supplying gutka to neighboring states via border cities like Bidar and Bengaluru (Fig 3).

Since tobacco is a profitable venture, our mapping of different supply chain actors demonstrated an integrated network—bound by tacit information sharing practices and access to infrastructure—where people with varied roles enter the chain on shared economic interests. The supply chain is made up of processes that operate independently but cooperatively. Traders were involved in procuring, manufacturing, storing and transporting as well as buying and re-selling gutka to vendors. Traders and vendors reportedly had close links with shopkeepers and sold exclusively to known customers. Shopkeepers frequently interacted with retailers and wholesale traders, while wholesalers supplied gutka to shopkeepers. Additionally, shopkeepers displayed gutka on purpose to lure customers into buying cigarettes and paan (betel leaf). Our observations are in line with results from a study conducted in Nepal, Bangladesh, and Pakistan that emphasizes the complexity and non-linearity of SLT supply chain management systems [49]. It is possible that this non-linearity, in conjunction with a variety of other tobacco products, allowed supply chain actors to adapt to external shocks such as the ban without compromising personal gains. A compliance study conducted in Mumbai found that consumer base and vendor profits were unaffected as alternatives like cigarettes and non-gutka SLT products were freely sold during the gutka ban [50].

Combating the use of SLT is slowly but surely becoming a policy priority in Southeast Asia, with countries such as Nepal, Bhutan, and Bangladesh amending existing tobacco control laws to prohibit SLT in public places, ban manufacturing and supply, and advertising. Simultaneously, multi-pronged strategies such as cessation services, pictorial warnings, taxation, education and awareness, vendor licensing, and litigation measures are being implemented [31, 51, 52]. Our study adds significant value and can assist policymakers, researchers, and practitioners better understand the gutka ban policy measure in India. Further investigation into the illicit supply-chain routes, enforcement timings, and complex social networks can empower tobacco control advocates and implementers in devising evidence-based interventions to ensure that the WHO-FCTC supply reduction measures are effectively implemented. Training and capacity building of science-communicators, journalists, and media firms can enable them to better represent public health interests by documenting and exposing industry interference. Long-term sustained media advocacy can contribute to the denormalization of SLT. High-burden countries can use media content analysis to identify local contexts in which tobacco market operates to fully explain the implementation progress.

### Strengths and limitations

Our study has several strengths. We examined news media sourced from Google search engine which has the largest database of reports available in the public domain. The mixed-method features of content analysis was useful in describing coverage patterns as well as interpret news stories [53]. Categories were drawn from the data as we read, re-read and familiarized ourselves with the news. An inductive approach allowed us to examine which arguments were reported, who had more influence, and how various stakeholders responded to the ban enforcement. In addition, a diverse research team of public health professionals, journalists, and social scientists facilitated cross-pollination of ideas, rich discussions, and critique resulting in a multidisciplinary body of work. Our study also encountered several methodological and procedural limitations. Based on the exploratory nature of the study, we used limited search terms. Despite this, the endless results produced by Google were overwhelming. We had insufficient knowledge about how Google search algorithms work to ensure replication of results. To minimize this, we restricted our searches to the first five pages. Manual search activities such as browsing history as well as differences in spelling (gutka vs gutkha), search terms (gutka vs smokeless tobacco) and place-based results mean that searches in different locations would yield different results, making generalizability of findings a challenge. Our analysis focused on English language news and excluded regional languages. As a result, ban enforcement coverage in some high-burden states like Tripura, Manipur, Odisha, Assam, and Arunachal Pradesh were inadequately captured. Furthermore, we did not include audio-visual content in the analysis, which means that their effect on perception and recall were not explored. Future studies need to examine local and regional news and audio-visual content as they are the important primary sources of information for many states in India. Since we worked with a large sample of data, the content analysis exercise was time-consuming and exhaustive, especially during the thematic analysis and interpretation of results. Administering such a method increases the likelihood to make simplistic comparisons as well as errors based on the subjective experiences of researchers [53]. We made an effort to minimize researcher bias by having two co-authors code the data and conducting numerous team discussions to resolve disagreements at each stage of the research process.

## Conclusions

Media content analysis of a policy intervention such as the gutka ban has provided rich descriptions of the stakeholders involved, arguments reported in the news, and the channels and processes associated with the banned SLT product. News media is a useful advocacy tool for communicating SLT-related health hazards, documenting potential industry interferences, studying competing interests, and analyzing implementation progress. Policymakers and implementers in charge of tobacco control in India could benefit from investing in news media research and analysis by incorporating the findings into developing targeted strategies. This is possible through collaboration and sustained engagement with media firms. Future research should also prioritize media coverage in vernacular languages and study other tobacco control policy interventions to improve the overall reporting of SLT control in India.

## Supporting information

**S1 Table. News media reports included in the review.**
(PDF)

**S2 Table. Characteristics of the included news media reports.**
(PDF)

**S3 Table. State-wise quantity and amount worth of gutka seized.**
(PDF)

## Acknowledgments

We sincerely thank Riddhi Dsouza and Upendra Bhojani of the Chronic Conditions and Public Policies cluster at IPH Bengaluru for sharing their insights on tobacco control laws. We are grateful to colleagues Prashanth NS, Swathi SB, Shivanand Savatagi, Meena Putturaj, and Yogish CB who gave critical inputs at various stages during a capacity building write-shop conducted for early-career researchers at IPH Bengaluru. Special thanks to Rajesvari Parasa, independent researcher and friend, for offering her technical expertise in GIS.

## Author Contributions

**Conceptualization:** Vivek Dsouza, Pragati B. Hebbar.

**Data curation:** Vivek Dsouza, Pratiksha Mohan Kembhavi, Praveen Rao S., Kumaran P., Pragati B. Hebbar.

**Formal analysis:** Vivek Dsouza, Pratiksha Mohan Kembhavi, Pragati B. Hebbar.

**Funding acquisition:** Pragati B. Hebbar.

**Methodology:** Pragati B. Hebbar.

**Project administration:** Pragati B. Hebbar.

**Software:** Praveen Rao S.

**Supervision:** Pragati B. Hebbar.

**Validation:** Vivek Dsouza, Kumaran P., Pragati B. Hebbar.

**Visualization:** Vivek Dsouza.

**Writing – original draft:** Vivek Dsouza.

**Writing – review & editing:** Pratiksha Mohan Kembhavi, Praveen Rao S., Kumaran P., Pragati B. Hebbar.

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
