## [Decision Letter · Decision Letter 0]

14 Oct 2022

PGPH-D-22-00341

Enforcing smokeless tobacco control regulation in India: A content analysis of news media coverage on the gutka ban.

Dear Dr. Hebbar,

Thank you for submitting your manuscript to PLOS Global Public Health. After careful consideration, we feel that it has merit but does not fully meet PLOS Global Public Health’s publication criteria as it currently stands. Therefore, we invite you to submit a revised version of the manuscript that addresses the points raised during the review process.

In addition to addressing the comments from the two reviewers, please add a limitations section within the Discussion and explicitly discuss study limitations, including the methods used for media content analysis.

We look forward to receiving your revised manuscript.

Kind regards,

Madhukar Pai, MD, PhD

Editor-In-Chief

Journal Requirements:

1. Please send a completed 'Competing Interests' statement, including any COIs declared by your co-authors. If you have no competing interests to declare, please state "The authors have declared that no competing interests exist". Otherwise please declare all competing interests beginning with the statement "I have read the journal's policy and the authors of this manuscript have the following competing interests:"

a. State what role the funders took in the study. If the funders had no role in your study, please state: “The funders had no role in study design, data collection and analysis, decision to publish, or preparation of the manuscript.”

b. If any authors received a salary from any of your funders, please state which authors and which funders.

Reviewers' comments:

Reviewer's Responses to Questions

**Comments to the Author**

1. Does this manuscript meet PLOS Global Public Health’s publication criteria? Is the manuscript technically sound, and do the data support the conclusions? The manuscript must describe methodologically and ethically rigorous research with conclusions that are appropriately drawn based on the data presented.

Reviewer #1: Yes

Reviewer #2: Yes

2. Has the statistical analysis been performed appropriately and rigorously?

Reviewer #1: Yes

Reviewer #2: N/A

3. Have the authors made all data underlying the findings in their manuscript fully available (please refer to the Data Availability Statement at the start of the manuscript PDF file)?

Reviewer #1: No

Reviewer #2: Yes

4. Is the manuscript presented in an intelligible fashion and written in standard English?

Reviewer #1: Yes

Reviewer #2: Yes

5. Review Comments to the Author

Reviewer #1: Dear authors,

My appreciations for taking up this vital study, with key findings. The study has been well drafted and the effort is commendable. The authors have done extensive literature work for the article and it is reflected in the quality of the manuscript. Except for a few minor revisions, the article is well-written.

Abstract

Although the objective states that the study aims to understand how the news shape public opinion, the results mentioned in the abstract does not reflect upon the impact on public opinion. I suggest the objective to be reworded.

Results

The result section can be minimised to just stating the outcomes of the study rather than adding any descriptions. All the description and explanation are to be added to the discussion part.

Pictorial or figure representation of the overall result might give an prompt idea of the major outcome of this study to the readers

Discussion

While there are ample results reported, the discussion feels inadequate. The authors can consider reducing the content on strengths and limitation, while discussion few more key components from results.

Figures

The figures are nowhere cited in the main text, which makes it difficult to find the relevance of the figures to the manuscript.

Reviewer #2: Thank you for this important/useful analysis and novel approach in understanding the implementation of tobacco control laws. I have some minor suggestions to strengthen the paper and connect it to the wider literature.

In the introduction section it will be good to mention that

a) India has ratified WHO FCTC in 2004, which has specific articles (especially 9 and 10) that aim to reduce the impact of SLT use. And it has exceeded the recommended provisions and enforced a ban considering the health impact of SLTs.

b) That this ban was a result of multisectoral coordination between NGOs, Supreme Court and policy makers.

Here is a WHO source for reference: (p13) and please use other sources as needed

https://apps.who.int/iris/bitstream/handle/10665/363378/9789240052291-eng.pdf

Mentioning this background info will connect this article to wider FCTC literature/be easily discoverable in searches and provide better context.

In the methods section:

Mention the location (state/country) from where the search was conducted in Google – as Google search results are impacted by location. The fact that you conducted this search from India will be a strength.

Discussion:

Suggest including a sentence or two about other countries that have banned SLT in the region as you speak about LMICs maybe around line 410.

Line 409: Suggest rephrasing to something like “As combating the use of SLT is becoming a policy priority” the current wording is a little unclear.

Line 415: I think it is good to say WHO FCTC than WHO – as they provide guidance on supply reduction etc. and is separate from WHO

6. PLOS authors have the option to publish the peer review history of their article (what does this mean?). If published, this will include your full peer review and any attached files.

**Do you want your identity to be public for this peer review?** For information about this choice, including consent withdrawal, please see our Privacy Policy.

Reviewer #1: **Yes: **Revathy Sudhakar

Reviewer #2: No

---

## [Decision Letter · Decision Letter 1]

24 Feb 2023

How does Indian news media report smokeless tobacco control? a content analysis of the gutka ban enforcement.

PGPH-D-22-00341R1

Dear Dr Hebbar,

We are pleased to inform you that your manuscript 'How does Indian news media report smokeless tobacco control? a content analysis of the gutka ban enforcement.' has been provisionally accepted for publication in PLOS Global Public Health.

Best regards,

Julia Robinson

Executive Editor

Reviewer Comments (if any, and for reference):

Reviewer's Responses to Questions

**Comments to the Author**

1. If the authors have adequately addressed your comments raised in a previous round of review and you feel that this manuscript is now acceptable for publication, you may indicate that here to bypass the “Comments to the Author” section, enter your conflict of interest statement in the “Confidential to Editor” section, and submit your "Accept" recommendation.

Reviewer #1: All comments have been addressed

Reviewer #2: All comments have been addressed

2. Does this manuscript meet PLOS Global Public Health’s publication criteria? Is the manuscript technically sound, and do the data support the conclusions? The manuscript must describe methodologically and ethically rigorous research with conclusions that are appropriately drawn based on the data presented.

Reviewer #1: Yes

Reviewer #2: Yes

3. Has the statistical analysis been performed appropriately and rigorously?

Reviewer #1: Yes

Reviewer #2: Yes

4. Have the authors made all data underlying the findings in their manuscript fully available (please refer to the Data Availability Statement at the start of the manuscript PDF file)?

Reviewer #1: Yes

Reviewer #2: Yes

5. Is the manuscript presented in an intelligible fashion and written in standard English?

Reviewer #1: Yes

Reviewer #2: Yes

6. Review Comments to the Author

Reviewer #1: The authors have addressed all the queries raised and made appropriate revisions. The manuscript may be considered for acceptance and publication

Reviewer #2: Thanks for meticulously addressing the comments.

7. PLOS authors have the option to publish the peer review history of their article (what does this mean?). If published, this will include your full peer review and any attached files.

**Do you want your identity to be public for this peer review?** For information about this choice, including consent withdrawal, please see our Privacy Policy.

Reviewer #1: **Yes: **Revathy

Reviewer #2: No
